# Facile Method for Surface-Grafted Chitooligosaccharide on Medical Segmented Poly(ester-urethane) Film to Improve Surface Biocompatibility

**DOI:** 10.3390/membranes11010037

**Published:** 2021-01-04

**Authors:** Yifan Liu, Zhengqi Liu, Ya Gao, Weiwei Gao, Zhaosheng Hou, Yuzheng Zhu

**Affiliations:** 1College of Chemistry, Chemical Engineering and Materials Science, Shandong Normal University, Jinan 250014, China; xiaofanfan34@163.com (Y.L.); liuzhengqi6618@163.com (Z.L.); kira.omg@icloud.com (Y.G.); 2Sagacis Biotechnology Co., Ltd., Jinan 250101, China; gww1512@163.com; 3Shandong Tianming Pharmaceutical Co., Ltd., Jinan 250100, China; yuzhengzhu22@163.com

**Keywords:** segmented poly(ester-urethane), chitooligosaccharide, surface grafting, biocompatibility

## Abstract

In the paper, the chitooligosaccharide (CHO) was surface-grafted on the medical segmented poly(ester-urethane) (SPU) film by a facile two-step procedure to improve the surface biocompatibility. By chemical treatment of SPU film with hexamethylene diisocyanate under mild reaction condition, free -NCO groups were first introduced on the surface with high grafting density, which were then coupled with -NH_2_ groups of CHO to immobilize CHO on the SPU surface (SPU-CHO). The CHO-covered surface was characterized by FT-IR and water contact angle test. Due to the hydrophilicity of CHO, the SPU-CHO possessed higher surface hydrophilicity and faster hydrolytic degradation rate than blank SPU. The almost overlapping stress-strain curves of SPU and SPU-CHO films demonstrated that the chemical treatments had little destruction on the intrinsic properties of the substrate. In addition, the significant inhibition of platelet adhesion and protein adsorption on CHO-covered surface endowed SPU-CHO an outstanding surface biocompatibility (especially blood compatibility). These results indicated that the CHO-grafted SPU was a promising candidate as blood-contacting biomaterial for biomedical applications.

## 1. Introduction

Segmented polyurethanes (SPUs) have been extensively used as biomaterials because of their superior tensile properties and adequate biocompatibility [1,2,3,4]. They are not only used in the manufacturing of chronic implants, but also have promise for bone implants and orthopedic due to the suitable processability, high flexibility, and bio-adhesion properties [5,6,7,8]. However, inflammatory mechanisms and wound healing could be elicited by the existence of a foreign matter in a physiological environment [9]. The main reason for the failure of implantable medical devices is the quick aggregation of proteins on the material surface and the occurrence of subsequent responses, such as platelet coagulation and activation [10]. Nowadays, improving the biocompatibility of SPUs is a widespread concern. Surface modification methods, especially surface grafting, have been investigated to prevent biofouling and polish up the biocompatibility for biomedical applications [11,12,13,14]. By producing a non-fouling surface coating, the surface can be protected from the nonspecific protein coating to make the device “stealthy”. Natural biopolymers will not induce any toxicity or adverse response because of their excellent biocompatibility. In addition, they are biodegradable and can be metabolized completely after use [15,16]. Thus, the natural biopolymers are developed as biomaterials to replace or restore a function of the body while in contact with the body fluids.

Chitosan (CH), a cationic polysaccharide found in crustacean shells, is obtained by partial deacetylation of chitin. Chitooligosaccharide (CHO), derived from CH via hydrolysis, has a smaller molecular size, lower viscosity, and better solubility than CH in water [17]. CHO has been known to possess various biological activities, such as anti-inflammation, anti-tumor, anti-oxidative and anti-microbial abilities [18,19,20,21]. Clearly, the CHO-modified SPUs can improve the performances (especially biocompatibility) of medical SPUs, but the relevant researches are rarely reported. In our previous paper [22], a new kind of medical polyurethane was prepared with CHO as a cross-linking agent. The biodegradable CHO-based polyurethanes exhibited excellent tensile properties and good surface blood compatibility, showing great potential as a long-term implantable biomaterial. 

In the present work, the CHO was surface-grafted on the segmented poly(ester-urethane) (SPU) film contained well-defined hard segments via a facile two-step procedure to improve the surface biocompatibility without destroying the intrinsic tensile properties of the substrate. The SPU film was firstly treated with small molecule diisocyanate to fix free –NCO groups on the surface with high grafting density, which were then coupled with –NH_2_ of CHO to immobilize CHO on the surface. The film surface of SPU before and after modification were characterized, and the tensile properties, in vitro degradability, surface hydrophilicity and water absorption were investigated. Moreover, platelets adhesion and protein adsorption were used to evaluate the surface biocompatibility.

## 2. Materials and Methods

### 2.1. Materials

CHO (*M*_n_: 3000 g/moL) was offered by Shandong Hailongyuan Biotechnology Co., Ltd. (Weifang, China), and the degree of deacetylation was about 93.5%. Poly(ε-caprolactone) (PCL, *M*_n_ = 2000 g/moL), stannous octoate (Sn(Oct)_2_) and hexamethylene diisocyanate (HMDI) were supplied from Sigma-Aldrich (St. Louis, MO USA). Before use, the CHO and PCL were dehydrated thoroughly at 90 °C for 8 h under vacuum. Toluene, obtained from Aladdin Reagent Co. (Beijing, China), was dried with P_2_O_5_ and re-distilled under vacuum. Phosphate buffered saline (PBS, pH = 7.4) came from Shanghai Yuanmu Biological Technology Co., Ltd., Shanghai, China.

According to our previous publish paper [23], the biodegradable SPU was prepared by reaction of PCL with well-defined diurethane diisocyanate (HBH) in the presence of Sn(Oct)_2_, and the SPU film with a thickness of 0.3 ± 0.03 mm, which was acquired via solvent evaporation. The chemical structure was confirmed by ^1^H NMR and FT-IR, and the molecular weight was measured by GPC (*M*_w_ = 109,000, *M*_n_ = 80,500, *M*_w_/*M*_n_ = 1.35).

### 2.2. Surface-Grafting of CHO onto the SPU Film

CHO was surface-grafted onto the SPU film by a two-step procedure, as shown in Figure 1. HMDI (2.04 g) and Sn(Oct)_2_ (0.02 g) were dissolved in anhydrous toluene (20 mL) to obtain a homogeneous solution. The SPU film (2 × 2 cm^2^) was immersed in the solution and the system was shaken at 25 °C for 48 h. After that, the film was removed and soaked in anhydrous toluene with continuous shaking for an additional 30 min, then completely washed with anhydrous toluene to obtain the SPU film with surface-grafted –NCO groups (SPU-NCO). The SPU-NCO films were put into the 20 mL aqueous solution containing 2.0 g CHO. After shaking (50 rpm) at 16 °C for 180 min, the film was removed from the solution, first rinsed with deionized water to detach the unreacted CHO and then with ethanol to remove the water. The obtained SPU film with surface-grafted CHO (SPU-CHO) was dried under vacuum to a constant weight.

### 2.3. Instruments and Characterization

FT-IR spectra were obtained on a Nicolet-560 spectrophotometer (Nicol, Madison, WI, USA) with a scanning range of 400 to 4000 cm^−1^.

Tensile properties were investigated by a universal tensile machine (Shenzhen Regal Instruments Co., Ltd., Shenzhen, China) according to the standard test method in GB/T1040-2006 (China).

Surface hydrophilicity was characterized by water contact angle (WCA) method using a CAM 200 contact angle goniometer (KSV Instruments Ltd., Helsinki, Finland). The images were collected at fixed interval from the deposition of the water drop to 30 s. 

The water uptake (WUP) was used to assess the bulk hydrophilicity of a material. The dry film sample (*M*_d_) was incubated with deionized water at 37 °C, and at 6 h intervals, the sample was weighted (*M*_s_) after removing the surplus water on the surface with filter paper. The WUP was expressed as the weight percentage of water in the sample: WUP (%) = (*M*_s_ − *M*_d_)/*M*_d_ × 100.

In vitro degradability of SPU and SPU-CHO films were quantified through the weight loss in PBS solution [24]. The film sample was weighed (*W*_o_) and placed into 10 mL PBS (pH = 7.4). After different incubation time, the sample was rinsed with deionized water, dried under vacuum at 35 °C and weighted (*W*_d_). The weight loss was obtained as the following equation: Weight loss (%) = (*W*_o_ − *W*_r_)/*W*_o_ × 100. 

Platelet-rich plasma (PRP), which was acquired from fresh rabbit blood by centrifugation [25], was used to assess the platelet adhesion on the film surface. After being equilibrated with PBS, the film sample (1 × 1 cm^2^) was soaked with 1.5 mL PRP for 1 h at 37 °C. The film was fetched out and washed with PBS. The adhered platelets were immobilized by immerging the film in 2.5% glutaraldehyde for 0.5 h at 37 °C. Afterwards, the sample was rinsed with PBS and dehydrated by using a series of ethanol-water solutions (50, 60, 70, 80, 90, 100% (*v/v*)). The morphologies of the platelet-adhered surface were visualized using a JSM-5400 scanning electron microscope (SEM) (JEOL, Tokyo, Japan) with a beam energy of 5 kV and a working distance of 8.4 mm.

The bovine serum albumin (BSA) was used to detect the quality of adsorbed protein on the film surface [26]. The film sample was incubated in BSA solution (1.0 mL, 45 µg/mL) for 1 h at 37 °C. After the film sample was completely washed with PBS, the surface-absorbed protein was detached with sodium dodecyl sulfonate aqueous solutions (SDS, 1 *wt*%). The concentration of adsorbed protein was measured by a micro-Bradford protein assay kit (LMAI Biol., Shanghai, China).

## 3. Results and Discussion

### 3.1. FT-IR

The blank and modified surface was characterized by FT-IR, and the results of SPU, SPU-NCO, CHO (powder) and SPU-CHO films are shown in Figure 2. The representative absorption peaks observed at 3317, 1720, 1689, and 1150 cm^−1^ in the spectrum of SPU film (Figure 2a) were assigned to the characteristic absorption of –NH, amide I, amide II and ester bond C–O–C stretching vibration, respectively [27]. After SPU film was treated with HMDI, a new absorption peak at about 2269 cm^−1^ appeared (Figure 2b), which was attributed to the characteristic absorption of -NCO stretching vibration, strongly supporting that –NCO groups had been successfully introduced onto the film surface. Although the reaction was performed under mild reaction conditions, the surface-grafting density of –NCO groups, determined with n-butylamine titration [28], was 5.23 × 10^−7^ mol/cm^2^, which was much higher than polyurethanes based on diphenylmethane diisocyanate (2.5 × 10^−8^ mol/cm^2^) [29]. The high grafting density was closely related to the high –NH– content in well-defined hard segments, which provided more reactive points.

The surface-grafted –NCO groups were reacted with the –NH_2_ groups of CHO to immobilize CHO on the SPU surface. In order to introduce more CHO on the surface, excess CHO was used and the reaction was performed at low temperature of 16 °C without catalyst. Thus, the side reaction of neighboring –NCO on the surface reacting with –NH_2_ in one CHO molecule was minimized. Thus, the measured relative mass gain (8.9 ± 0.75 *wt*%), obtained by comparing the dry weight of SPU and SPU-CHO samples, was only slightly lower than the calculated value (11.2 *wt*%). In addition, as shown in Figure 2d, the absorption peak at 2269 cm^−1^ vanished, alongside the appearance of the absorption peak at 1031 cm^−1^, which corresponded to the cyclic ether of CHO (Figure 2c). The results indicated that the -NCO on the film surface reacted completely and the CHO was introduced to the surface.

### 3.2. Mechanical Properties

In order to evaluate the influence of chemical treatments on the substrate, the mechanical properties of the films before and after modification were measured. The typical stress-strain curves of SPU and SPU-CHO films are displayed in Figure 3, and tensile properties obtained from the curves are listed in Table 1. The non-modified SPU exhibited outstanding mechanical properties with a breaking stress of 30.2 MPa, an ultimate elongation of 948%, and a storage modulus of 19.1 MPa, which was ascribed to its compact H-bond-linking network structure [30]. After the CHO was grafted on the surface, only a slight reduction in the values of tensile properties was discovered (Table 1). The stress-strain curve of SPU-CHO film was almost coincided with that of blank SPU film manifested that the chemical treatments had little influence on the mechanical properties of the substrate. Obviously, the excellent mechanical properties of SPU-CHO are likely to be more suitable for some special implants.

### 3.3. Surface Hydrophilicity and Water Absorption

The surface biocompatibility was affected by surface hydrophilicity to a certain degree [31]. The surface hydrophilicity of the SPU and SPU-CHO films were investigated by a WCA test [32,33], and the time dependence of WCA is exhibited in Figure 4. The blank SPU had a hydrophobic surface with a high WCA of 87.4°, while the WCA on the CHO covered surface (SPU-CHO) decreased to 23.8°, indicating that the surface hydrophilicity was improved substantially. In additional, the WCA on SPU surface needed much shorter time to reach the steady value than that on SPU-CHO surface. It should be ascribed to the hydrophilic hydroxyl groups and unreacted amine groups of CHO exposed on the surface, which could bind water molecule to form a hydration layer through hydrogen bonds. It was further evidence that the CHO was grafted onto the surface, which was conducive to improve the surface biocompatibility.

In polyether-based PU, the hydrolytic degradability was closely related to the bulk hydrophilicity, which could be reflected directly by WUP. Generally, the high WUP meant a fast hydrolytic degradation rate. Figure 5 shows the curves about the time dependence of the films WUP. From Figure 5, it could be seen that the SPU-CHO film possessed much higher saturated WUP (9.6%) than the SPU film (1.35%). Moreover, the modified film needed more time (48 h) to reach saturated WUP than the blank film (24 h). Obviously, the CHO grafted on the surface could bond a certain amount of water, and the ordered arrangement of polymer structure on the surface was inevitably destroyed by the chemical treatment, which prompted the water molecules to get into the substrate.

### 3.4. In Vitro Degradation

Considering the temporary biomedical application, the implant biomaterials should possess biodegradability. PBS (pH = 7.4) was selected to simulate the physiological microenvironment, in which the in vitro degradation of blank and modified films was carried out to assess the influence of surface-grafted CHO on the biodegradability of the substrate. The data expressed the percentage weight loss of the SPU and SPU-CHO films with time are exhibited in Figure 6. The blank SPU film showed a very slow degradation rate with less than 2% weight loss after six months of degradation and only 11% weight loss until the end of the measurement. While the SPU-CHO film displayed 18.5% weight loss at 6 months, and the film lost it original shape and became fragments at 10 months with a weight loss of 45%. The result indicated that the degradation rate of SPU-CHO film was much higher than that of the blank SPU film. It was mainly ascribed to the hydrophilic surface-grafted CHO, which facilitated the water molecules to approach the substrate and attack the ester groups easily. The results were consistent with the water absorption described above. From the weight loss curves of in vitro degradation, the surface-grafted hydrophilic CHO was beneficial to improve the hydrolytic degradation of the substrate.

### 3.5. Platelet Adhesion

With regards to blood-contacting materials, blood compatibility is one of the most important biocompatibilities [31]. It is generally known that, when an artificial material comes into contact with blood, it results in the platelet adhesion, making the coagulation pathways active, and then facilitating thrombosis [34]. Thus, the surface platelet adhesion on the films in PBS was carried out to evaluate the blood compatibility. The typical SEM images of the SPU and SPU-CHO films are presented in Figure 7. Massive platelets adhered on the surface of the blank SPU film (Figure 7a), and part of the platelets presented shape variation. Compared with the SPU film, the SPU-CHO surface showed a significantly lower adhesion of platelets (Figure 7b), which indicated that the surface has effective inhibition of platelet adhesion—that is, better blood compatibility.

### 3.6. Protein Adsorption

Biocompatibility of the implant material is also associated with protein adsorption because the surface-adsorbed proteins can trigger the coagulation sequence [35]. The protein adsorption response on the material surface plays a significant role in designing the biomedical devices [36]. The BSA was chosen to evaluate the protein absorption, and the quantity of absorbed BSA protein on the blank and CHO-covered surface is shown in Figure 8. Compared with the bare surface (SPU: 11.46 µg/cm^2^), there was a great decrease in BSA absorption on the CHO-covered surface (SPU-CHO: 2.98 µg/cm^2^). Obviously, the hydration layer formed on the film CHO-covered surface could produce repulsive force to protein and reduce the interaction between the protein and film surface. It was known that the adsorption of protein on the surface of the implant materials was the first step in thrombus formation [37]. Hence, the low protein adsorption capacity meant good surface biocompatibility (especially blood compatibility), making SPU-CHO material a potential candidate for in vivo biomedical applications.

## 4. Conclusions

By a facile two-step procedure, the natural biopolymer of CHO was surface-grafted on the SPU film (SPU-CHO). The CHO-covered surface was characterized by FT-IR and the WCA test. Due to the hydrophilicity of CHO, the SPU-CHO possessed higher surface hydrophilicity and faster hydrolytic degradation rate than blank SPU. The almost overlapping stress-strain curves of SPU and SPU-CHO films demonstrated that the chemical treatments had little destruction on the intrinsic properties of the substrate. Furthermore, the significant inhibition of protein adsorption and platelet adhesion on CHO-covered surface endowed SPU-CHO an outstanding surface biocompatibility (especially blood compatibility). Consequently, the CHO-grafted SPU materials might be promising candidates for biomedical applications, although more detailed biological assessments must be carried out.

## Figures and Tables

**Figure 1 membranes-11-00037-f001:**
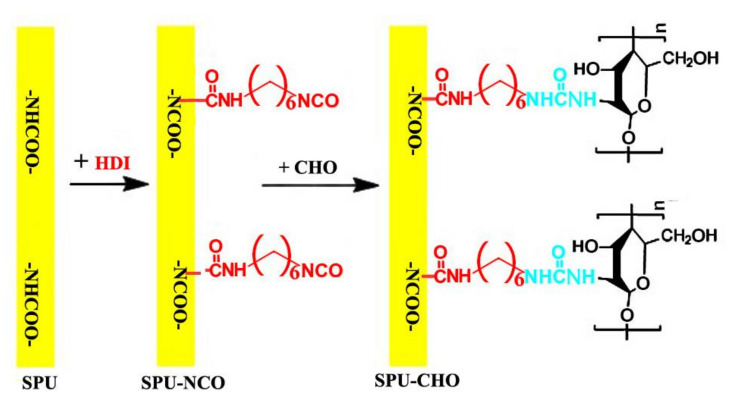
The surface-grafted CHO onto SPU film.

**Figure 2 membranes-11-00037-f002:**
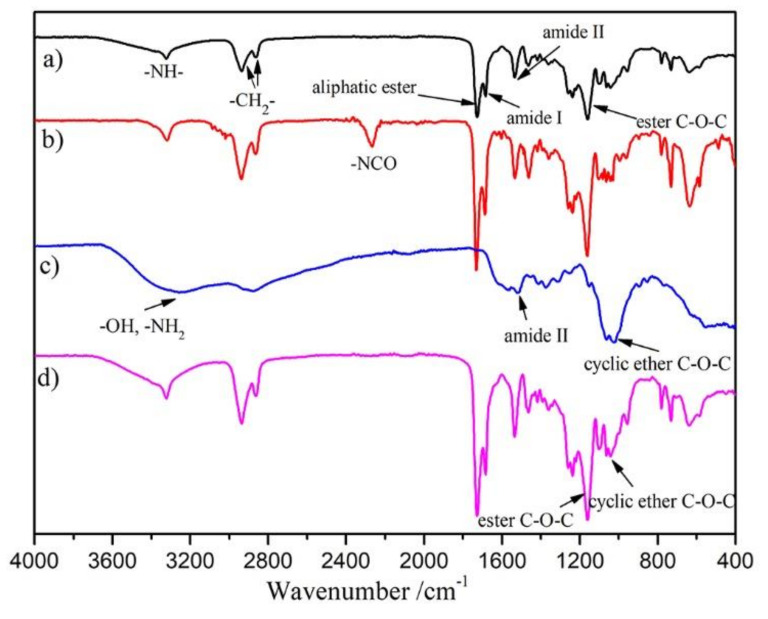
FT-IR spectra of (**a**) SPU, (**b**) SPU-NCO, (**c**) CHO (powder) and (**d**) SPU-CHO films.

**Figure 3 membranes-11-00037-f003:**
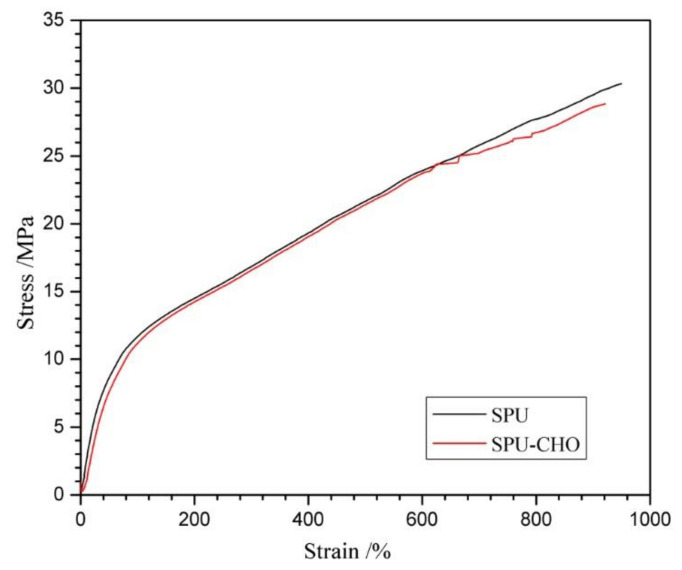
Typical stress vs. strain curves for SPU and SPU-CHO films.

**Figure 4 membranes-11-00037-f004:**
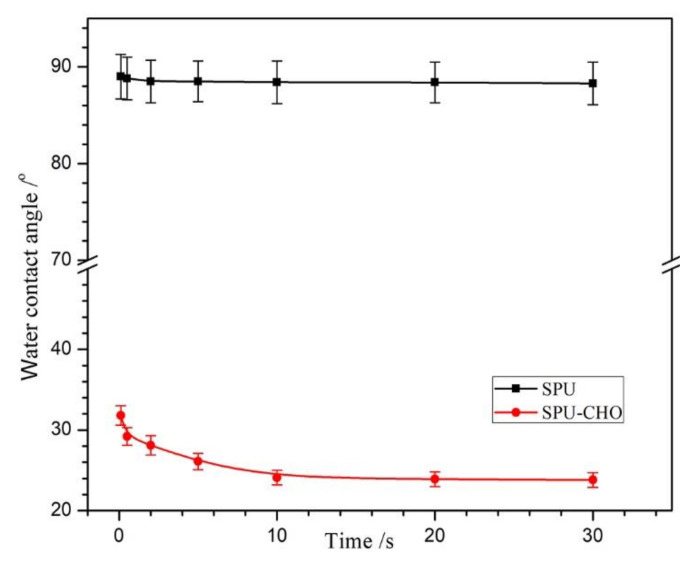
The time dependence of water contact angles on the SPU and SPU-CHO film surface (*n* = 5).

**Figure 5 membranes-11-00037-f005:**
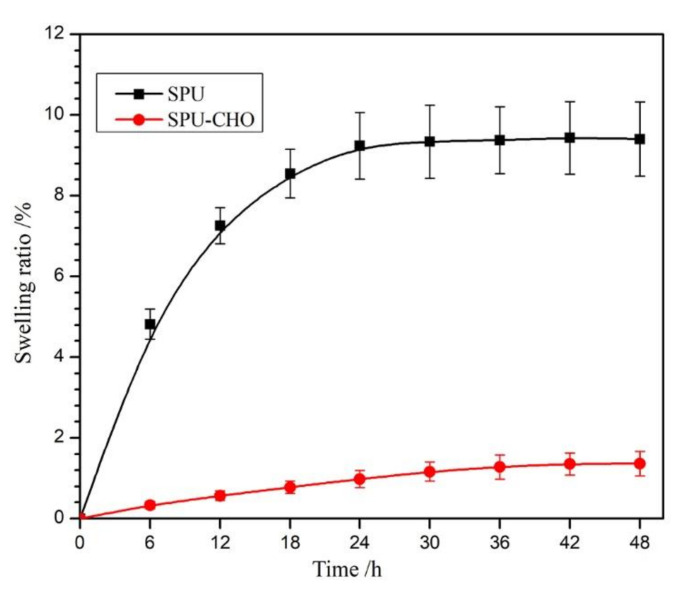
The time dependence of water uptake of SPU and SPU-CHO film at 37 °C in deionized water (*n* = 3).

**Figure 6 membranes-11-00037-f006:**
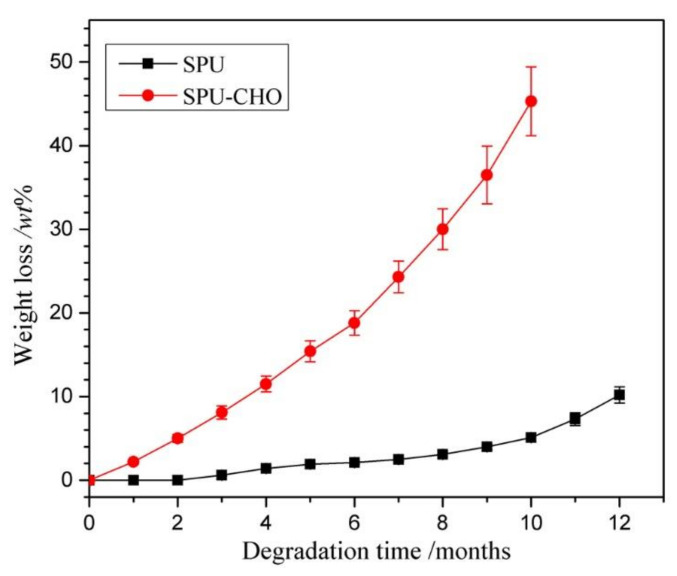
Degradation behaviors of the SPU and SPU-CHO films in PBS (*n* = 5).

**Figure 7 membranes-11-00037-f007:**
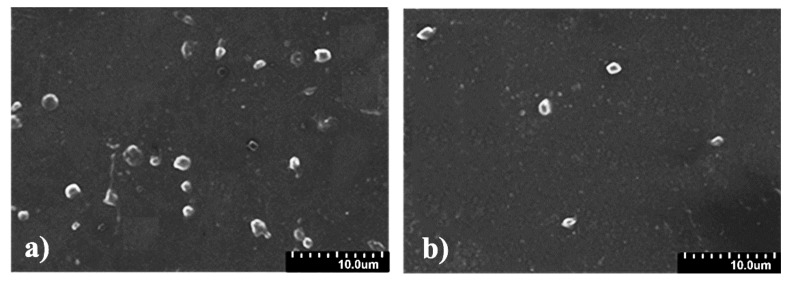
Typical SEM images of platelet adhesion on the film surface of (**a**) SPU and (**b**) SPU-CHO.

**Figure 8 membranes-11-00037-f008:**
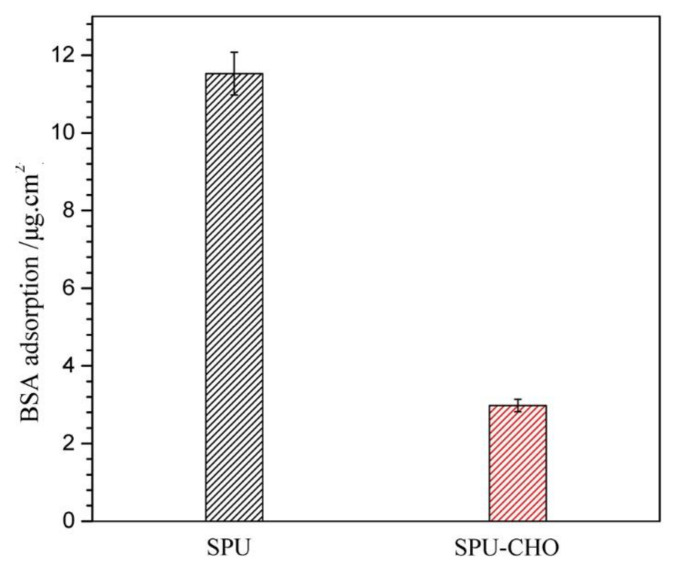
Quantity of absorbed BSA protein on the surface of SPU and SPU-CHO films (*n* = 3).

**Table 1 membranes-11-00037-t001:** Tensile properties of SPU and SPU-CHO films (*n* = 5).

Films	Breaking Stress*/*MPa	Ultimate Elongation/%	Storage Modulus/MPa
SPU	30.2 ± 1.6	948 ± 38	19.1 ± 1.2
SPU-CHO	28.8 ± 1.4	922 ± 35	17.8 ± 1.0

## Data Availability

Not applicable.

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
