# Peer review of "Facile Method for Surface-Grafted Chitooligosaccharide on Medical Segmented Poly(ester-urethane) Film to Improve Surface Biocompatibility"

_membranes, 2021, doi:10.3390/membranes11010037_

Round 1

Reviewer 1 Report

Review

The paper entitled ‘Facile method for surface-grafted chitooligosaccharide on medical segmented poly(ester-urethane) film to improve surface biocompatibility’ is quite interesting research work which concerns on medical based PU modified by CHO. However this work needs revision because of lack of information in the experimental part.

Below you can find some comments:

  1. Add to the FTIR analysis (fig.2) spectra of chitooligosaccharide. It should improve to notice the changes in the structure and fully confirm  the well performed grafting process.
  2. In the manuscript very little analysis of the repeatability and reproducibility of measures are addressed. In the experimental work performed, how many samples were tested for each study? Please add.
  3. How did you calculate the amount of CHO, optimum for grafting process?? It should be explained.
  4. Did you check another quantity of CHO for SPU surface grafting??
  5. Have you checked the surface hydrophilicity and water absorption of CHO?? Should be also analyzed.

Reviewer 2 Report

The paper is focused on the fabrication of biocompatible films by surface grafting on poly(ester-urethane). The topic falls within the scope of the journal. I recommend the publication after the following revisions:

  • I suggest to determine the stored energy up to breaking by the analysis of stress vs strain curves.
  • Contact angle measurements. Did the authors monitor the time dependence of the water contact angle? As reported in literature for biocompatible films with variable structure [New J. Chem., 2018,42, 8384-8390; Carb. Polym. 2017, 170, 198–205], the analysis of the water contact angle vs time trends provide the kinetic constant and the exponential parameter, which is related to the absorption/spreading contributions. Both parameters can be very useful to investigate the effect of the morphology on the wettability properties of the films.
  • The scale length within SEM images is not clear. Please check and revise.
  • Experimental details (energy of beam, working distance,...) for SEM measurements should be added.

Round 2

Reviewer 1 Report

The manuscript after revision reached new quality. Nevertheless, in the corrected manuscript Authors forgot to attached Fig.2.

Author Response

Response to Reviewer 1

Nevertheless, in the corrected manuscript Authors forgot to attached Fig.2.

Response: we are very sorry for our negligence, and the Fig.2 was attached in the revised manuscript.

Reviewer 2 Report

The paper was improved according to the reviwers' comments. I recommend its publication in the current form.

Author Response

Thanks for your approval.